# *Fusarium* Toxins in Chinese Wheat since the 1980s

**DOI:** 10.3390/toxins11050248

**Published:** 2019-04-30

**Authors:** Jianbo Qiu, Jianhong Xu, Jianrong Shi

**Affiliations:** 1Key Laboratory for Control Technology and Standard for Agro-product Safety and Quality, Ministry of Agriculture and Rural Affairs/Jiangsu Key Laboratory for Food Quality and Safety-State Key Laboratory Cultivation Base, Ministry of Science and Technology/Collaborative Innovation Center for Modern Grain Circulation and Safety/Institute of Food Safety and Nutrition, Jiangsu Academy of Agricultural Sciences, Nanjing 210014, China; qiujianbo19850901@126.com; 2School of Food and Biological Engineering, Jiangsu University, Zhenjiang 212013, China

**Keywords:** China, wheat, *Fusarium* species, mycotoxicoses, *Fusarium* toxins, toxin management

## Abstract

Wheat Fusarium head blight (FHB), caused by *Fusarium* species, is a widespread and destructive fungal disease. In addition to the substantial yield and revenue losses, diseased grains are often contaminated with *Fusarium* mycotoxins, making them unsuitable for human consumption or use as animal feed. As a vital food and feed ingredient in China, the quality and safety of wheat and its products have gained growing attention from consumers, producers, scientists, and policymakers. This review supplies detailed data about the occurrence of *Fusarium* toxins and related intoxications from the 1980s to the present. Despite the serious situation of toxin contamination in wheat, the concentration of toxins in flour is usually lower than that in raw materials, and food-poisoning incidents have been considerably reduced. Much work has been conducted on every phase of toxin production and wheat circulation by scientific researchers. Regulations for maximum contamination limits have been established in recent years and play a substantial role in ensuring the stability of the national economy and people’s livelihoods.

## 1. Introduction

Wheat (*Triticum aestivum* L.), which belongs to the grass family, is widely distributed throughout the world and has a large planting area and total yield. Wheat is not only a vital nutrient-rich food source but also an important industrial raw material and animal feed component, and its distribution and use in China is similar to its global characteristics. With the adjustment of the agricultural structure, wheat has been the third largest food crop after maize and rice. According to the relevant data from the 2018 China Statistical Yearbook, wheat acreage has stabilized at 245,080 km^2^ over the past ten years, accounting for 14.73% of cereal production in 2017. China’s total production of wheat has reached 134 million tons and has increased 22% from 2007. However, despite the increase, the deficit in the trade balance increased to a billion dollars, which means that China needs to import approximately 4.4 million tons of wheat every year; this importation makes it imperative to breed wheat for increased yield. China is the largest wheat producer, supplying 17% of the total yield globally. More importantly, China is also the largest wheat consumer and accounts for 16% of the total consumption of wheat every year [1]. Wheat flour-based products, such as steamed buns and noodles, are staple foods for more than half of China’s population and can be traced back thousands of years. Thus, assuring the quality and safety of wheat and its products is vitally significant to the national economy and people’s livelihoods.

Like other crops, due to the influence of agricultural environments and inputs, there are various safety issues, such as heavy metals and pesticide residues, in wheat-derived products, [2,3]. According to previous studies, mycotoxins are an even greater challenge for wheat quality [4,5,6]. Fusarium head blight (FHB), or scab, mainly caused by members of the *Fusarium graminearum* species complex (FGSC), is a destructive fungal disease in most of the world’s major wheat-growing areas [7]. From an economic perspective, FHB seriously affects wheat yield, resulting in income reductions for farmers and a substantial financial loss to society. Recently, the impact of FHB on food safety and human health has aroused considerable public concern because the diseased grains are frequently contaminated with toxic secondary metabolites. Among these, type B trichothecenes and zearalenone (ZEN) are the most serious and prevalent toxins in China. Trichothecenes, mainly deoxynivalenol (DON), 3-acetyl-deoxynivalenol (3ADON), 15-acetyl-deoxynivalenol (15ADON), and nivalenol (NIV), can cause acute poisoning symptoms, such as vomiting and dizziness in humans and weight loss and anorexia in animals. The mechanism of action of trichothecenes is based on the inhibition of protein synthesis [8]. Additionally, DON presents immunotoxicity, cytotoxicity, teratogenicity, and carcinogenicity [9,10,11,12]. ZEN and its metabolites have strong estrogenic activities and can cause reproductive alterations [13]. ZEN is also reported to be hepatotoxic, hematotoxic, immunotoxic and genotoxic [14]. Therefore, even low levels of these toxins in raw grains can make them hazardous to human or animal health.

Concerning the possible substantial effect of toxins on the country’s economy and society, the Chinese government has made many efforts aimed at ensuring cereal safety, such as setting strict limit and rigorous analytical method standards, revising agricultural product quality security laws, and establishing national special projects for agro-product safety risk evaluations. *Fusarium* toxins are produced by *Fusarium* species in suitable environmental conditions, and once they were present, there is no effective way to completely eliminate them from food. Even so, agricultural scientists have developed physical, chemical and biological methods to interfere with *Fusarium* species infection and toxin accumulation in all phases of wheat production. The present review aims to systematically analyze the occurrence characteristics of the main *Fusarium* toxins in wheat, the varying trend in human poisoning incidents, and the utility of toxin management measures in the past 40 years to provide a reference for scientific monitoring and effective prevention and control of mycotoxin contamination in China.

## 2. Human Mycotoxicoses Caused by *Fusarium* Toxins

Wheat flour and its products have always accounted for a large proportion of the traditional Chinese diet and have a long history, especially in the north of the country. Due to the widespread occurrence and the complicated toxic effects of poisonous FHB metabolites, their presence in food should be regarded as a potential food safety hazard. Mycotoxicosis outbreaks in humans have been associated with the intake of food contaminated with these toxins and are thus a matter of great concern to consumers.

Historically, there were several reports about mycotoxicoses in humans after the consumption of scabby wheat grains, particularly in the 1980s. A total of 21 outbreaks of human intoxication occurred successively from 1984 to 1991 in the main wheat-growing provinces [15]. Some cases of scabby wheat-related food poisoning published in professional journals are listed in Table 1. The typical clinical symptoms are gastrointestinal disorders, including abdominal pain and fullness, nausea, diarrhea, vomiting, fatigue, and fever.

In 1985, a severe FHB epidemic occurred in wheat in a suburb of Puyang County, Henan Province. A large proportion of local inhabitants showed typical symptoms including gastrointestinal disorders and nervous system disturbances after eating scabby wheat grains, and the incidence of related diseases was 45% [17]. Most of the patients returned to health after two hours of symptoms. After this incident, DON was considered the most important causative toxicant responsible for human intoxications.

After a large flood in 1991, extensive outbreaks of human toxicoses affecting a large number of people who consumed moldy wheat grains occurred in parts of Henan, Anhui, and Jiangsu provinces, as appropriate conditions, such as heavy rainfall and low temperature, for an FHB epidemic and mycotoxin production occurred after the flood [22,23,24]. High amounts of DON were found in food that was linked to an episode of ’red-mold intoxication’ involving 130,000 people with acute gastrointestinal disorders in Anhui Province in 1991 [29].

In the 21st century, human poisoning caused by *Fusarium* toxins decreased and moved from group outbreaks to individual incidents. For example, in 2000, a whole family in Northeast China developed acute poisoning symptoms twice after eating products derived from flour from home-grown wheat [27]. The latest large-scale food safety incident reported in Northwest China was demonstrated to be associated with the consumption of flour products processed from diseased wheat grains. In 2003, of 5043 people in 10 villages who consumed contaminated foods, 701 (13.9%, no lethal cases) presented intoxication. The most common symptoms of the victims were nausea, numbness of the throat, and a burning sensation of the esophagus; symptoms disappeared within approximately 3 hours without medical attention. This human food toxicity incident happened in June and July, when large quantities of wheat grains became moldy but were still harvested despite inferior quality resulting from long-term precipitation. Other reasons for this large-scale incident were a lack of health awareness and poor traditional eating habits of the local residents [28].

In addition to acute toxicity, *Fusarium* toxins are associated with some human chronic diseases (Table 2). Kashin-Beck disease (KBD) is a chronic, endemic osteochondropathy that occurs mainly in the distant provinces of northeastern and northwestern China [30]. In KBD-endemic areas, wheat has been frequently infected with *Fusarium* species, and the disease has been suspected to be associated with the consumption of *Fusarium*-contaminated grains. Earlier findings by Luo et al. [31] on the level of *Fusarium* toxins in wheat from Shanxi and Inner Mongolia found that the prevalence rate and content of DON 3ADON, NIV, and ZEN were significantly higher in the high-risk area of KBD than in the low-risk area. It seemed that there was a dose-effect relationship between toxins contamination level and degree of KBD occurrence. In Qinghai Province, the DON content in the wheat flour samples from the KBD area was significantly higher than that of the non-KBD area, and the authors indicated that the characteristic distribution of DON contamination in wheat flour was consistent with the KBD prevalence [32]. Similar results regarding DON contamination were reported in wheat flour from Shandong and Gansu provinces [33]. However, the precise etiology of KBD was still not clearly defined, and T-2 toxin and selenium levels were implicated as important causes. Fortunately, a recent epidemiological investigation reported that the KBD prevalence decreased from 1990 to 2007 [34].

During the process of researching the etiology of esophageal cancer, DON was found to be one of the predominating contaminating mycotoxins in the grains and foodstuffs in the high-incidence area in China [39]. Co-contamination of DON, 15ADON, and NIV in wheat samples from Henan Province, where cancer is highly prevalent, was studied in 1989, 1995 and 1997 [36,37]. Linxian samples showed higher levels of DON than those from Shangqiu, which has a low esophageal cancer incidence. These studies suggested that fungal contamination of foods and residents’ exposure to mycotoxins could be considered possible factors involved in the development of cancer. In a survey of Shandong samples harvested in 2010, significantly higher levels of DON were identified in all wheat samples from high-risk areas than in wheat samples from low-risk areas [38]. Hsia et al. [39] reported that NIV naturally existed at high levels in dietary food in high-risk cancer areas and suggested that people who consumed a diet with high levels of NIV had a significantly greater risk of developing esophageal cancer than those who consumed a diet with low levels of NIV. The occurrence of esophageal cancer is a result of multiple factors, including the poor eating habits (tobacco smoking and alcohol drinking), the lack of nutrients and trace elements, genetic determinants, and the intake of food containing high concentration of nitrite compounds and mycotoxins. It is speculated that the chronic consumption of cereals contaminated with mycotoxigenic *Fusarium* species can increase the odds of the disease, although no correlation between *Fusarium* toxins and esophageal cancer has been published [40]. Nonetheless, as the most important natural poisons, the threat should not be taken lightly. In general, most outbreaks of human mycotoxicoses from *Fusarium* toxins followed the occurrence of severe FHB. In the last few years, the damage caused by FHB has become increasingly serious; however, the incidence and the number of victims of human mycotoxicoses have followed the opposite trend. Along with the improvement in living standards, moldy grains are no longer used for food, but they can still serve as major feed components for poultry and livestock. This harm to the development of the animal husbandry industry is not mentioned here, yet it needs to be solved urgently. There are three main causes for this phenomenon. The first is the development and popularization of science and technology. Consumers have a remarkable increase in the understanding and knowledge of scabby wheat and *Fusarium* toxins, thus contributing to the development of good eating habits. Another reason is the improvement of Chinese living standards. People have more dietary choices and can discard diseased or low-quality cereals. The regulatory authorities for all levels of agricultural product quality and safety also play a crucial role. Moldy wheat grains have been eliminated from the market, and the quality of wheat products is under strict surveillance; thus, safe and superior quality foods are provided to consumers. All these factors reduce the risk of the occurrence of food safety incidents.

## 3. Toxin-Producing *Fusarium* Strains

The first FHB outbreak in China can be traced back to 1936; scientists began to research the pathogen of this disease approximately 20 years later [41] and found that most strains from China can produce large amounts of DON and ZEN [42,43]. Based on relevant research in recent years, *F. graminearum* was identified as the main pathogen of FHB in China, despite the presence of a large variety of *Fusarium* species isolated from diseased wheat grains. As early as the 1980s, research on FHB pathogens in 21 provincial regions in China identified 18 *Fusarium* species, among which *F*. *graminearum* dominated and accounted for 94.5% of the total [44]. Subsequently, similar results were obtained from studies in Henan, Fujian, Hunan, Ningxia and Qinghai provinces [45,46,47,48].

In recent years, genetic and molecular approaches have led to a new understanding that narrowly defines *F. graminearum* sensu stricto (s. str.) as a species complex with significant genetic diversity, clear divergence of biological species, and an obvious geographical distribution. Currently, FGSC consists of at least 16 phylogenetically distinct species [49]. *F. graminearum* s. str. is the most widely distributed species and occurs in most FHB areas around the world [50], while *F. asiaticum* is the main FHB pathogen present in Asia [51,52].

FGSC species can produce several mycotoxins; type B trichothecenes are the most common toxic metabolites found in infected cereals [53]. FGSC strains usually present one of three trichothecene profiles: (i) deoxynivalenol and 3-acetyldeoxynivalenol (3ADON chemotype); (ii) deoxynivalenol and 15-acetyldeoxynivalenol (15ADON chemotype); or (iii) nivalenol and its acetylated derivatives (NIV chemotype) [54]. The chemotype composition appears to be species dependent [55]. In China, most of the *F. graminearum* s. str. isolates are 15ADON producers, while *F. asiaticum* isolates contain 3ADON and NIV chemotypes. Table 3 presents the trichothecene type compositions of *F. graminearum* s. str. and *F. asiaticum* in wheat.

In terms of geographic distribution, the vast majority of *F. asiaticum* isolates have been collected from warm southern areas [56,57], and *F. graminearum* s. str. is mainly distributed in cool northern regions [58,59]. The composition of the populations has been stable over time. The population structure and genetic variation in FHB pathogens have been studied in detail in Jiangsu Province, which has a long history of rice growing and a large rice-growing area that covers 30 million acres. The *F. asiaticum* strain that produced 3ADON has always dominated [60,61], and no similar evidence of temporal trends in the North American wheat population or Chinese barley population has been found [62,63,64,65,66]. Extensive wheat-rice rotation is critical for *F. asiaticum* overwintering, and perithecium production typically favors rice straw under warmer conditions. The better fitness of *F. asiaticum* on rice and DON producers on wheat has led to the prevalence of 3ADON-producing *F. asiaticum* in most wheat-rice rotation areas in Southern China. As a result, a 3ADON-producing *F. asiaticum* population may have been present in this region for a long time, and we suggest that this might have been the main factor underlying the absence of variation in trichothecene genotype frequencies from 1976 to 2014. Recently, based on available data from the literature, we concluded that a cropping system with wheat/maize rotation selects for *F. graminearum*, while a wheat/rice rotation selects for *F. asiaticum* [67].

Due to the promotion and application of straw-returning methods, more perithecia form and more ascospores are released in the subsequent year. Crop debris in the field feeds and increases the amount of primary FHB inoculum. Climatic or agricultural conditions favor wheat infection by *Fusarium*, which eventually leads to higher levels of toxins in wheat. A dominant FHB population means that the main toxins have been substantially retained over a long period of time, and the quantity of FHB pathogens is likely to continually increase. Based on this phenomenon, it can be assumed that the epidemic risk of FHB and harmful levels of *Fusarium* toxins could continue to increase for a long time in the future.

## 4. Natural Occurrence of *Fusarium* Toxins in Wheat

China is a traditional agricultural country, and wheat is an important food ingredient. FHB severity has exhibited an increasing trend; therefore, the investigation of and surveillance for *Fusarium* toxins in wheat and its products are of great significance. Table 4 and Table 5 show the main *Fusarium* toxins in wheat grains and wheat flour, respectively.

*Fusarium* toxin contamination shows various regional differences. The mean content and standard-exceeded rate of toxin was the highest in wheat and flour samples from Anhui and Jiangsu provinces, where FHB occurred severely and frequently. During the wheat heading and flowering period, a high temperature and humidity environment in the middle and lower reaches of the Yangtze River provided favorable conditions for the propagation of *Fusarium*, the occurrence of the disease and the accumulation of toxins. Although recent trends have indicated that FHB is spreading towards Northeast and Northwest China, the wheat quality in these regions is high. For example, a minimal incidence and concentration of DON and ZEN was detected in wheat from Heilongjiang [59]. A recent study reported the prevalence and concentration of DON in wheat harvested during 2013 from the northwest regions of China, suggesting varied and low levels of DON contamination in the region [77]. Especially in Xinjiang and Tibet, the safety quality of wheat and flour is in good condition [77,78,79]. These findings suggest that cold or drought environment conditions might be unfavorable for toxin production. Even so, some limitations, such as small sample sizes and deficient research, cannot be ignored.

*Fusarium* toxin contamination shows clear temporal dynamics; the epidemiologic degree of FHB has a clear relationship with the toxin contamination level. The occurrences of *Fusarium* toxins were studied in detail in Jiangsu, Anhui, and Henan provinces over a long period of time, and there was high concentration of *Fusarium* toxins over the standard rate in 1985, 1989, 1991, 2010, 2012, and 2015. This is an interesting discovery, as these unusual years were reported to have massive outbreaks of FHB, except for 1991, in which a large flood occurred. In 1985, Henan Province had the worst outbreak in 40 years; almost all the wheat grains were contaminated with DON, NIV, and ZEN in high amounts. The contamination rate for samples containing toxins higher than the tolerance limit of 1000 µg/kg was close to 50%, and the highest number of DON-positive samples reached 40,000.0 µg/kg [80]. In flour samples derived from wheat collected in 1989, the frequency of DON detection approached 100%, and the average DON content was 1334.0 and 577.7 µg/kg in Anhui and Jiangsu, respectively [81,82]. There was an exceedingly high rate (81.5%) of corresponding contamination in wheat, with a maximum concentration of 13,300.0 µg/kg [83]. In the last decade, the FHB epidemic has become more frequent and severe; moreover, toxin contamination, particularly by DON, is more common and serious. In 2010, 2012, and 2015, the average amounts of DON in Anhui wheat samples were 2701.0, 4501.6, and 17,753.8 µg/kg, respectively [84,85,86]. The amounts in Jiangsu wheat samples in the corresponding years were >1000.0, >3000.0, and >2000.0 µg/kg, respectively [85,87,88,89]. FHB outbreaks aggravate toxin contamination despite the inconspicuous effect on ZEN accumulation.

Recently, masked DON, a derivative of DON, has become a focus of attention for in-depth research. D3G (deoxynivalenol-3-glucoside) is a primary type of masked DON and is the most studied. D3G was for the first time detected in naturally contaminated maize and wheat in 2005 [90]. Until now, the toxicological research about D3G is rare. As the protein synthesis inhibitor, the activity of D3G was much lower than that of DON [91]. Although there was no direct evidence that D3G was more toxic than its precursor, the existing research data showed that D3G could release DON by a hydrolysis reaction in the metabolic process [92,93,94]. Similar studies have raised concerns about the metabolites of D3G in humans or animals and the occurrence of this toxin in wheat grains and their products. A total of 192 wheat samples from 2007–2008 collected in 7 provinces were analyzed for D3G accumulation, and the toxin was found in 52.0% wheat samples with an average content of 43.0 μg/kg [95]. In a recent study, high incidence rates and levels of D3G were detected in wheat samples from Jiangsu and Anhui provinces in 2015–2016. A total of 96.3% of wheat from Jiangsu was positive for D3G, with contamination rates ranging from 12.0 to 18061.0 μg/kg [89], while 99.5% of wheat from Anhui was contaminated by D3G, with contamination rates ranging from 28.3 to 2957.2 μg/kg [86]. The detection rate of D3G in wheat flour was to a certain degree, but the contamination level was obviously reduced. D3G was detected in 104/125 flour samples from 12 provinces (0.1–52.8 μg/kg) [96] and 38/158 wheat flour samples from 5 provinces (8.7–33.3 μg/kg) [97]. In 2010, 33.4% of Shandong wheat flour samples, 30.8% of Hubei wheat flour samples, and 5.46% of Hebei wheat flour samples contained D3G with average contamination rates of 1.1, <20.0, and 1.9 μg/kg, respectively [98,99,100]. During wheat processing, D3G was transferred to prepared products, such as bread and noodles, and caused toxic effects by producing DON after contact with intestinal digestive enzymes [101]. Thus, the fact that masked *Fusarium* toxins are also a potential risk to human health cannot be ignored.

From several national surveys, it was reported that multitoxin, mainly DON and ZEN, were widespread and severe in Chinese wheat grains, especially in the past decade. Although the overall contamination situation varied significantly in different years, wheat quality safety conditions increased in severity with the prevalence of FHB. The detection rate of *Fusarium* toxins in wheat flour remains at a certain level; nevertheless, most flour samples have toxin levels below Chinese regulatory limits and those of wheat grains. During wheat cleaning and flour milling, proper processing removes some toxins and is an effective measure in reducing toxin contamination. Considering the frequency and degree of the FHB epidemic, prolonged, successive, and extensive monitoring of *Fusarium* toxins in wheat and its products is essential for ensuring food safety and promoting human health.

## 5. *Fusarium* Toxin Management

Since FHB can lead to economic losses and health concerns, some comprehensive strategies for the reduction in the occurrence of FHB and *Fusarium* mycotoxins need to be developed. A combination of planting resistant cultivars, adapting agricultural practices, and applying chemical and biological controls for reducing fungi invasion and toxin production may help to control the occurrence of FHB and its associated mycotoxins.

### 5.1. Expansion of the Basic Knowledge about Toxin Production

Similar to pigments and antibiotics, *Fusarium* toxins are secondary metabolites produced by *Fusarium* species during its natural growth; these metabolites are closely associated with cell differentiation, growth, and the response to the external environment. Successful whole genome sequencing of several *Fusarium* species has provided useful data for researchers to study the biosynthesis pathways and regulatory mechanisms of mycotoxins [128] and has led to the complete resolution of the trichothecene gene cluster (TRI-cluster) [129,130] and zearalenone biosynthetic gene cluster [131,132]. Environmental factors, including pH [133,134], carbon sources [135], nitrogen sources [136,137], H_2_O_2_ [138], availability of free water (a_w_), incubation temperatures [139] and regulatory signaling pathways, including the mitogen-activated protein kinase (MAPK) [140], cyclic adenosine phosphate-protein kinase A (cAMP–PKA) [141,142], and target of rapamycin (TOR) pathways [143], can influence toxin accumulation. These molecular mechanisms offer targets for the inhibition of DON synthesis by genetic engineering technology.

Several studies on detailed species and chemotype identification, population genetic diversity, and biological characteristics of FHB pathogens from wheat in China have been performed [60,66,144]. 3ADON-producing populations with high toxin accumulation are more dominant. The identification of advantageous populations, temporal dynamics evolution trends and ecological adaptations could provide a theoretical basis for confirming a regulatory focus, breeding for FHB resistance, and developing targeted disease and mycotoxin control strategies.

### 5.2. Maturation of Chemical Control Measure of FHB

Spraying carbendazim during the wheat flowering period is a common method for controlling FHB and has played a crucial role in integrated disease control programs since the 1970s. However, the monitoring results of *Fusarium* toxins in wheat samples calls into question the true effects of chemical control on toxin production. In the past, control was measured only by the decrease in the visual disease index and the recovery of economic losses; serious toxin contamination was largely ignored. As such, the efficacy of carbendazim on toxin accumulation is questionable. Poor reductions in toxin accumulation may be due to the increase in *Fusarium* species that are resistant to carbendazim and the improved synthesis ability of DON in resistant strains compared with that of conventional strains [145,146]. High doses of the fungicide and the high proportion of resistant strains in the field may be associated with the serious toxin contamination in wheat samples from Jiangsu and Anhui provinces. In some regions with low carbendazim resistance, this compound can still reduce the disease index, diseased kernel rate and DON content remarkably [147].

Phenamacril, a new type of acrylate fungicide, was developed by China; phenamacril provides successful control of a variety of plant diseases caused by *Fusarium*. Field experiments indicated that phenamacril showed better efficacy against FHB than carbendazim, and the yield increase effect was equivalent to that of carbendazim [146]. Moreover, the application of phenamacril reduced the total DON level in wheat grains by more than 80% compared with the untreated controls, and phenamacril exhibited great potential in toxin management [147,148]. Currently, this fungicide has been popularized and applied in most wheat-growing areas.

Another kind of fungicide that can significantly reduce the ability of pathogens to infect plants and synthesize toxins is the azole antifungals, including metconazole, propiconazole, prothioconazole, and tebuconazole, which belong to the class of demethylation inhibitors. Several studies have proved that tebuconazole alone at varied concentrations or in combination with other fungicides was effective in inhibiting FHB and DON, and the drug showed optimal performance at the early anthesis stage or later anthesis stage [147,148,149,150]. The sensitivities of FHB pathogens to metconazole and the efficacy of this fungicide in FHB and DON control in China were reported in a recent study; metconazole exhibited better efficacy than phenamacril and carbendazim [151]. Propiconazole also had a certain role in toxin management [148], but it was reported that this fungicide may have the opposite effect on toxin control at low doses. Sublethal doses of propiconazole triggered H_2_O_2_ production in vitro and further induced DON accumulation in the pathogen [152].

According to a new study, validamycin, a type of fungicide for the control of crop diseases caused by *Rhizoctonia* species, had a remarkable effect on reducing DON synthesis by *F. graminearum* [153,154]. Further research on the molecular mechanism revealed that validamycin decreased glucose production by targeting trehalase and blocking the glycolytic pathway, thereby reducing pyruvate and DON production [153]. This finding provides us with important reference values and guidance in the screening of appropriate fungicides against harmful metabolites, as no fungicidal activity of validamycin against *F. graminearum* was discovered in vitro.

There are many agricultural practices to combat pathogens, diseases, and contaminants; however, chemical control is still the most powerful and effective measure. The effect of fungicides on mycotoxins is complicated and can be influenced by various factors, such as toxin-producing fungus, environmental conditions, and treatment dosage. More research on the mechanisms of toxin production and compound mode of action is needed to provide theoretical support for the scientific and reasonable use of fungicides. Chinese government is vigorously implementing the strategy of the dosage reduction and efficiency increase of chemical pesticides, as pesticide residue is a threat to food safety and cannot be ignored. More rational and efficient application of these substances is extremely essential.

### 5.3. Development of Process Control Technology

Wheat can experience a serious loss of tissue composition after infection with scab, resulting in a change in the wheat kernel appearance. Diseased grains usually have shriveled surfaces, declined particle diameters, decreased hardness, and decreased weight; therefore, specific gravity separation is the best method to remove infected wheat kernels before storage and processing. Historically, wind force was used to separate diseased and normal grains based on simple devices, and notable results were obtained. Liang et al. [155] suggested that DON content could be effectively controlled by removing scabby grains with wind and sifting. More recently, a specific gravity separator combined with winnower has played an important role in wheat cleaning and purification. Li et al. [156] conducted particle size and specific gravity separation of raw material and found the redistribution of DON samples. After the elimination of impurities and infected grains, edible wheat was obtained with a 68.94% decrease in total DON compared with the level of DON in the original samples. Bian et al. [157] and Zhu et al. [158] further improved the conventional gravity separation technology; the mass fraction for the DON-contaminated fraction increased while the proportion decreased, indicating that the improved technology effectively and efficiently discarded DON-contaminated wheat grains.

It is well known that scabby wheat kernels present different shades of red due to the infection of *Fusarium*, and electric color sorting technology based on color characteristics can separate the contaminated grains to effectively manage toxins. The soft independent modelling of class analogies (SIMCA) model based on near infrared spectrum and hyperspectral imaging technology has been successfully applied in the identification of scabby grains and DON levels, with a recognition accuracy above 90% [159,160,161]. Shen et al. [162] established quantitative models for DON with attenuated total reflectance-Fourier transform infrared spectroscopy (ATR-FTIR) and developed a method for the rapid determination of DON in wheat and its products by associating the absorption values of samples with various DON contents with different bands. With the combined gravity and color sorting technology and equipment in the cleaning process, the DON content decreased from 1.56 mg/kg to 1.18 mg/kg [163].

Cleaning is the first step in wheat processing and plays a vital role in wheat flour safety. At present, color and gravity sorters have been implemented in most flour milling enterprises and are of great importance to mycotoxin reduction and the quality improvement of wheat flour and its products.

### 5.4. Improvement of the Standard System

To protect public health from the negative impacts of *Fusarium* toxins, many countries and food safety regulators, including China, have introduced maximum or recommended toxin levels for food and feed. Specific regulations at the national level are released by authoritative bodies (Table 6). The former National Health and Family Planning Commission (NHFPC) and China Food and Drug Administration (CFDA) jointly issued national food safety standards (GB 2761-2017), in which the main mycotoxin limits of food were specified. The regulatory limits of mycotoxins in the revised national food safety standards in the infant formula (GB 10765), older infant formula (GB10766), young children (GB 10767), and raw milk (GB 19301-2010) general rules for aged-food (draft for comments) were determined to be in accordance with GB 2761. The former General Administration of Quality Supervision, Inspection and Quarantine of the People’s Republic of China (AQSIQ) and Standardization Administration of China (SAC) promulgated hygienic standards for feed (GB 13078) and fully stipulated limit values for toxicants in feedstuffs and products. These standards help the government protect the of health citizens through food safety guarantees. As technical barriers in international trade, they may also contribute to the protection of domestic markets and industry development.

The national standard system of mycotoxin detection has been improved (Table 7), and some detection standards provide technical support for agricultural product quality and safety supervision, particularly for the most advanced analytical instruments, such high-performance liquid chromatography (HPLC), ultraperformance liquid chromatography (UPLC), and high-performance liquid chromatography-tandem mass spectrometry (HPLC-MS/MS). This progression reflects the level of technological innovation in the application and promotion of standards.

However, safety and detection standard systems for agriculture products are in some developing countries and regions are still incomplete. China focuses mostly on primary agricultural products, and agro-products from varied processes for different purposes and consumer bases lack specific standards. For instance, there are no specific rules for direct consumables. Moreover, the process of establishing standards is slightly behind. China does not have established maximum levels for D3G, NIV, and other common toxins. As for minor crops, most lack limitation standards and related testing standards. Combined contamination of multitoxin is a common feature of cereal grains, and a standard for the simultaneous determination of multitoxin is a top research priority.

## 6. Conclusions and Challenges for the Future

In the present review, we gathered available Chinese data from the last half century on the occurrence of *Fusarium* species and toxins in wheat as well as the resultant food-poisoning incidents. Because of climatic conditions and cropping systems, there are increased amounts of infection sources and greater risks of FHB epidemics and *Fusarium* toxin contamination. Fortunately, with the advancement of society and the development of new materials, the human mycotoxicosis incidence has decreased gradually.

*Fusarium* toxin contamination in cereals remains an inevitable problem worldwide. In recent years, risk assessment and mycotoxin monitoring of cereals has been strengthened, and research in related fields has been promoted. However, achieving the goal of effectively controlling mycotoxin contamination in cereals is still a long way away. Integrated mycotoxin management practices, including preharvest control (e.g., tillage and crop rotation, selection of resistant varieties, proper sowing dates and density, irrigation and fertilization regimes, weed elimination, insect management, and chemical and biological control), harvest control (e.g., proper harvest time, professional mechanical equipment, mechanical damage reduction, effective cleaning, and impurity removal), and postharvest control (e.g., timely and efficient drying, good storage practices, and classified applications) should be employed to manage all possible risk factors to prevent mycotoxin contamination. In the future, with the effective implementation of good agricultural practices (GAPs), good manufacturing practices (GMPs), and hazard analysis critical control points (HACCPs), food safety, and consumer health can be improved and guaranteed as much as possible.

## Figures and Tables

**Table 1 toxins-11-00248-t001:** Outbreaks of moldy wheat-related food-poisoning incidents in humans since the 1980s.

Year	Region	Consumer	Victim	Reference
1985	Lingtao, Gansu	1549	1351	[16]
1985	Puyang, Henan	217	101	[17]
1988	Yulin, Guangxi	160	40	[18,19]
1988	Tongshan, Jiangsu	9	9	[20]
1989	Zigong, Sichuan	7	7	[21]
1989	Baihe, Shannxi	5016	701	[21]
1991	Xincai, Henan	840	479	[22]
1991	Linquan, Anhui	93	67	[23]
1991	Fuyang, Anhui	354	263	[24]
1991	Xuyi, Jiangsu	141	117	[24]
1996	Putian, Fujian	3	2	[25]
2000	Kedong, Helongjiang	6	6	[26]
2003	Dongming, Shandong	4	4	[27]
2003	Baihe, Shannxi	5043	701	[28]

**Table 2 toxins-11-00248-t002:** Occurrence of *Fusarium* toxins in high KBD or cancer incidence regions.

Year	Regions	Sample Size	Toxin	Incidence (%)	Average Content (µg/kg)	Content (µg/kg)	Reference
1989	Shannxi (KBD high incidence)	5	DON	100.0	514.0	343.0–1051.0	[31]
			3ADON	100.0	363.0	15.0–731.0	
			NIV	100.0	183.0	17.0–373.0	
			ZEN	40.0	15.0	5.0–25.0	
	Shannxi (KBD low incidence)	5	DON	100.0	184.0	73.0–410.0	
			3ADON	0	0	0	
			NIV	60.0	10.0	8.0–13.0	
			ZEN	0	0	0	
	Inner Mongolia (KBD high incidence)	6	DON	100.0	101.0		
			3ADON	50.0	24.0		
			NIV	33.3	9.0		
			ZEN	0	0	0	
	Inner Mongolia (KBD low incidence)	7	DON	100.0	75.0		
			3ADON	28.6	57.0		
			NIV	0	0	0	
			ZEN	0	0	0	
2010	Qinghai (KBD high incidence)	23	DON	87.0	302.0		[32]
	Qinghai (non-KBD)	27	DON	29.6	199.0		
2010	Gansu (KBD high incidence)	20	DON	75.0	142.8	12.8–205.3	[33]
		20	DON	80.0	137.4	28.5–180.7	
		20	DON	75.0	127.3	20.6–176.1	
	Shandong (KBD low incidence)	20	DON	10.0	12.9	10.2–15.6	
		20	DON	20.0	18.1	8.9–27.3	
1989	Henan Linxian (esophageal cancer high risk)	15	DON	46.7	59.0	7.0–309.0	[35]
		15	NIV	0	0	0	
	Henan Shangqiu (esophageal cancer low risk)	15	DON	46.7	18.0	7.0–36.0	
		15	NIV	46.7	15.0	13.0–21.0	
1995	Linxian, Henan (esophageal cancer high risk)	25	DON	92.0	83.0	9.0–193.0	[36,37]
			15ADON		0	0	
			NIV		29.0	13.0–50.0	
	Henan Shangqiu (esophageal cancer low risk)	15	DON	60.0	40.0	15.0–125.0	
			15ADON		0	0	
			NIV		12.0	4.0–22.0	
1997	Linxian, Henan (esophageal cancer high risk)	15	DON	66.7	28.0	0–138.0	
			15ADON		0	0	
			NIV		95.0		
	Henan Shangqiu (esophageal cancer low risk)	15	DON	0	0	0	
			15ADON		0	0	
			NIV		0	0	
2000	Linxian, Henan (esophageal cancer high risk)	9	DON	77.8	732.0	0–1614.0	[38]
		11	NIV	100.0	666.0	190.0–1476.0	
	Cixian, Hebei (esophageal cancer low risk)	18	DON	100.0	1031.0	176.0–4280.0	
		18	NIV	100.0	731.0	102.0–2105.0	
2010	Shandong (esophageal cancer high risk)	20	DON	100.0	195.2	70.1–302.8	[39]
		20		100.0	184.3	85.4–287.5	
	Shandong (esophageal cancer low risk)	20		10.0	12.9	10.2–15.6	
		20		20.0	18.1	18.9–27.3	

**Table 3 toxins-11-00248-t003:** The ratio of *F. graminearum* and F. asiaticum strains with varied chemotypes in partial studies in China.

Year	Region	Sample Sizes	Fg15ADON	Fg3ADON	FgNIV	Fa3ADON	FaNIV	Fa15ADON	Reference
1975–1980	China	2450	94.5%						[44]
1975–1981	Hunan	185	97.2%						[45]
1978–1981	Fujian	1081	99.1%						[46]
1985	Henan	241	98.0%						[47]
1985–1987	Ningxia	350	63.8%						[48]
1991–1992	Qinghai	27	56.5%						[68]
1993–1995	Qinghai	1005	54.3%						[69]
1999	China	299	22.7%			51.8%	17.7%	7.7%	[55]
2000	Zhejiang	208				42.3%	57.7%		[56]
2007–2014	Henan	327	89.0 %			6.7%	0.9%	0.3%	[70]
2008	China	444	38.1%			38.5%	21.9%	1.6%	[71]
2008	Sichuan	90	6.7%	3.3%	3.3%	25.56%	52.2%		[72]
	Chongqing	6	33.3%				66.7%		
	Hubei	201	12.4%	8.0%	0.5%	58.7%	4.5%	10.5%	
	Henan	25	100.0%						
	Anhui	42	45.2%	7.1%	4.8%	35.7%	7.1%		
	Jiangsu	69	15.9%	10.1%		63.8%	10.1%		
2008–2010	Jiangsu	292	5.5%			84.6%	9.9%		[73]
	Anhui	71	21.1%			59.2%	19.7%		
	Henan	88	89.8%			8.0%	2.3%		
	Hebei	23	100.0%						
	Shandong	31	83.8%			12.9%	3.2%		
	Hubei	25	92.0%				8.0%		
2008	Fujian	59					76.0%	24.0%	[57]
2009		100				4.0%	81.0%	15.0%	
2009	Hubei	168	9.5%			69.6%	6.6%	7.7%	[74]
2011–2012	Shandong	95	94.7%	4.4%	1.1%				[75]
2011	Jiangsu/Anhui	350	4.0%	4.0%		82.9%	4.3%	4.9%	[60]
2012	Jiangsu/Anhui	541	8.7%	1.5%		75.1%	9.4%	5.4%	
2013	Northeast China	118	64.4%						[58]
2014	Sichuan	103	18.5%			7.8%	71.8%	1.0%	[67]
	Hubei	57				87.7%	7.0%	5.3%	
	Anhui	93	1.1%			87.1%	10.8%	1.1%	
	Jiangsu	67	3.0%			89.6%	7.5%		
	Fujian	217	2.8%			10.6%	67.7%	18.9%	
2014–2015	Shandong	120	84.2%	4.2%	1.7%	10.0%			[76]
2016	Northeast China	84	44.1%	3.6%		1.2%	4.8%	22.6%	[59]

**Table 4 toxins-11-00248-t004:** Recent mycotoxin survey data in wheat grains in China.

Year	Regions	Sample Sizes	Toxins	Incidence (%)	Average Content (µg/kg)	Range (µg/kg)	Exceedance Rate (%)	Reference
1983	Anhui	40	DON	100.0	1161.4		22.5	[83]
1986		182		44.5	312.9		7.1	
1989		81		100.0	2640.0	0–13,300.0	81.5	
1991		26		100.0	2105.8		57.7	
1986	Anhui	150	DON	53.3	340.0	0–4000.0		[102]
		83	ZEN	22.9	32.0	0–300.0		
1991	Anhui	10	DON	100.0	15,900.0	2000.0–50,000.0	100.0	[22]
2007–2008	Anhui	25	D3G	64.0	45.5	2.2–238.4		[95]
			NIV	72.0	53.3	1.8–229.9		
			3ADON	44.0	6.3	1.8–18.4		
			15ADON	12.0	2.6	2.3–3.0		
			DON	100.0	46.3	3.7–169.3	0	
			ZEN	20.0	12.9	3.3–36.1	0	
2010	Anhui	21	DON	90.5	2701.0	521.0–4975.0	81.0	[84]
2012	Anhui	22	DON	95.4	4501.6	465.0–9930.0		[85]
2015	Anhui	370	DON	100.0	17,753.8	109.6–86,255.1		[86]
			D3G	99.5	414.4	28.3–2957.2		
			NIV	87.8	250.2	0–2399.7		
			3ADON	80.0	39.6	0–284.1		
			15ADON	67.3	13.2	0–184.7		
			ZEN	68.7	25.7	0–1091.4	5.1	
1986	Jiangsu	202	DON	26.7	40.0	0–400.0	0	[83]
		54	ZEN	64.8	51.0	0–300.0		
1991	Jiangsu	7	DON	71.4	2900.0	1560.0–5000.0		[23]
2007–2008	Jiangsu	24	D3G	62.5	59.3	1.7–179		[95]
			NIV	29.2	12.4	1.9–29.5		
			3ADON	29.2	5.5	2.1–11.3		
			15ADON	4.2	2.4	0–2.4		
			DON	95.8	73.0	2.8–408.3	0	
			ZEN	16.7	3.9	1.7–6.6	0	
2010	Jiangsu	35	DON	88.6	1221.0	259.0–3900.0	51.4	[84]
2010	Jiangsu	41	DON	100.0	1075.2	151.6–2550.2	44.0	[87]
			ZEN	46.3	216.0	10.1–3048.9	9.8	
2011		64	DON	32.8	82.1	14.5–1579.8	3.1	
			ZEN	0	0	0	0	
2012		75	DON	96.0	306.7	16.3–41157.1	48.0	
			ZEN	5.3	3.2	50.2–72.6	2.6	
2012	Jiangsu	62	DON	95.1	3260.9	260.0–11,200.0		[85]
2013	Jiangsu	66	ZEN	37.9	11.8	6.5–110.0		[88]
2014		66		46.9	22.0	15.0–194.3		
2015		70		54.3	39.3	25.1–307.3		
2015	Jiangsu	443	DON	100.0	2087.0	166.0–14,960.0		[89]
			D3G	96.0	545.0	83.0–5092.0		
2016		439	DON	100.0	2601.0	12.0–18,061.0		
			D3G	97.0	819.0	0–6708.0		
1985	Henan	19	DON	100.0	17,500.0	1.0–40,000.0	100.0	[17]
			ZEN	10.5	375.0	250.0–500.0	10.5	
1985	Henan	191	DON	99.0	923.0	15.9–3337.8	47.1	[80]
		191	NIV	81.0	128.2	12.5–608.6		
		92	ZEN	100.0	15.3	3.3–149.9		
1986	Henan	100	DON	74.0	14.2	6.7–175.4	0	
			NIV	4.0		9.5–49.0		
1986	Henan	97	DON	57.7	40.0	0–400.0	0	[102]
		60	ZEN	11.7	8.0	0–50.0	0	
1991	Henan	35	DON	100.0	1500.0	1000.0–3500.0	100.0	[23]
1991	Henan	24	DON			0-3500.0	41.7	[103]
1998	Henan	31	DON	96.8	2850.0	177.0–14,000.0		[29]
			15ADON	64.5	365.0	59.0–1800.0		
			NIV	3.2	578.0	0–578.0		
			ZEN	67.7	209.0	9.0–1400.0		
		28	DON	89.3	223.0	53.0–1240.0		
			15ADON	0	0	0		
			NIV	0	0	0		
			ZEN	25.0	108.0	1.0–217.0		
1999		34	DON	85.3	294.0	74.0–941.0	0	
			15ADON	0	0	0		
			NIV	0	0	0		
			ZEN	58.8	23.0	5.0–113.0		
2007–2008	Henan	28	D3G	75.0	24.6	2.8–171.1		[95]
			NIV	0	0	0		
			3ADON	14.3	3.6	0–4.9		
			15ADON	46.4	4.1	0–17.7		
			DON	100.0	74.6	2.9–363.6	0	
			ZEN	17.9	3.3	0–8.1	0	
1986	Shanghai	100	DON	100.0	340.0	0–2000.0		[102]
			ZEN	33.0	11.0	0–780.0		
1995	Shanghai	100	DON	53.0	280.9	0–1919.7	10.0	[104]
			NIV	35.0	103.4	0–1428.0		
2009–2012	Shanghai	198	DON	80.8	64.7	0.5–604.0	0	[105]
2011–2012	Shanghai	38	3ADON	100.0	10.3	0.7–35.2		
			15-DON	100.0	1.4	0.5–6.2		
2007–2008	Hebei	25	D3G	60.0	88.9	5.6–388.0		[95]
			NIV	24.0	12.3	1.8–57.5		
			3ADON	48.0	12.5	1.6–70.8		
			15ADON	88.0	236.1	1.5–1256.2		
			DON	80.0	167.3	1.7–636.2	0	
			ZEN	56.0	126.1	4.7–930.4		
1986	Gansu	135	DON	57.0	2050.0	0–20,000.0		[102]
		101	ZEN	40.6	15.0	0–300.0		
2013	Shannxi	81	DON	96.1	515.3	79.0–3030.0	8.64	[106]
	Ningxia	26		100.0	804.4	71.0–2330.0	26.92	
	Gansu	52		86.5	294.4	0–1798.0	1.92	
	Xinjiang	22		0	0	0	0	
2000–2016	Tibet	199	DON	0	0	0	0	[77]
			ZEN	0.5			0.5	
2007–2008	Sichuan	30	D3G	0	0	0		[95]
			NIV	73.3	17.7	3.0–39.1		
			3ADON	0	0	0		
			15ADON	0	0	0		
			DON	50.0	16.4	3.0–47.8	0	
			ZEN	20.0	5.1	2.0–8.8	0	
2007–2008	Chongqing	30	D3G	73.3	72.5	9.8–235.3		[95]
			NIV	100.0	199.5	7.8–1035.8		
			3ADON	60.0	9.1	1.9–34.6		
			15ADON	46.7	9.6	2.1–71.0		
			DON	100.0	133.3	12.0–590.7	0	
			ZEN	80.0	199.4	2.3–3425.1		
2016	Heilongjiang	55	DON	0	0	0	0	[59]
1984	China	29	DON	51.7	401.7	0–2450.0	6.9	[107]
			NIV	37.9	267.3	0–6644.0		
			ZEN	44.8	6.8	0–32.0	0	
2003	China	48	ZEN	100.0	98.0	0–470.0	72.9	[108]
2005	China	190	DON	66.3	50.0	0–612.7	0	[109]
2007	China	229	DON	38.0	73.9	0-600.8	0	[110]
			ZEN	16.0	1.6	0–72.4		
2008	China	41	DON	97.6	425.5		9.8	[111]
			ZEN	68.3	152.4		41.5	
2010–2013	China	681	DON	66.5	72.8	774.8–14276.0		[112]
			AcDON	66.8	74.7	797.7–14604.2		

**Table 5 toxins-11-00248-t005:** Recent mycotoxin survey data in wheat flour in China.

Year	Regions	Toxins	Sample Sizes	Incidence (%)	Average Content (µg/kg)	Range (µg/kg)	Exceedance Rate (%)	Reference
1983–1991	Anhui	DON	132	92.4	1065.6		40.9	[83]
1988	Hebei	DON	50	54.0	75.0	0–173.0	0	[113]
1988–1989	Anhui	DON	100	90.0	1008.6		43.0	[114]
1988–1989	Shanghai	DON	25	100.0	79.8		0	[115]
			25	80.0	58.1		0	
1989	Anhui	DON	84	100.0	1334.0		58.3	[81]
1989	Jiangsu	DON	50	96.0	577.7		18.0	[82]
1996	Shanghai	DON	30	86.7	101.2		0	[104]
		NIV		56.7	53.3			
2009	China (13 provinces)	DON	292	100.0	178.4	0.5–2995.1	1.7	[116]
		ZEN		53.4	5.1	0.3–55.0	0	
		NIV		88.4	8.1	0.3–218.2		
2010	China (12 provinces)	DON	125	96.8	179.0	0.1–1016.8	0.8	[96]
		3ADON		64.0	2.2	0.1–19.8		
		15ADON		95.2	4.2	0.1–25.5		
		D3G		83.2	10.1	0.1–52.8		
		NIV		86.4	10.3	0.1–76.5		
		ZEN		72.8	3.5	0.1–52	0	
2010	China (28 provinces)	DON			143.0		4.2	[117]
2011					147.0		2.2	
2012					658.0		20.7	
2013			5678	58.7	317.0	0–56,100	4.7	
2010–2013	China	DON	3848	71.7	126.0	218.0–6922.0		[112]
		AcDON	3860	71.8	91.5	219.2–6922.0		
2010-	Shandong	DON	359	97.2	84.3	0–825.9	0	[98]
		3ADON		11.1	0.1	0–3.6		
		15ADON		14.2	0.5	0–11.1		
		NIV		40.4	1.4	0–23.9		
		D3G		33.4	1.1	0–15.7		
		ZEN		0	0	0	0	
2010-	Hubei	DON	26	69.2	129.4	0–2133.2	3.9	[99]
		D3G		30.8	<20.0	0–252.4		
2010-	Hebei	DON	348	91.4	240.0	0–1129.0	0.6	[100]
		15ADON		34.2	1.9	0–6.0		
		NIV		16.4	3.2	0–19.1		
		ZEN		13.2	8.4	0–98.8	0.3	
		D3G		5.5	1.9	0–3.9		
		3ADON		2.1	3.2	0–2.6		
2011	Hebei	DON	31	16.1	137.0	2.4–639.0	0	[118]
		3ADON		6.4	0.7	0.6–0.8		
		15ADON		0	0			
2012		DON	348	91.4	240.0	11.5–1130.0	0.6	
		3ADON		34.2	1.9	1.1–6.0		
		15ADON		3.2	2.1	1.5–2.6		
2013		DON	293	99.6	156.0	6.2–878.0	0	
		3ADON		0	0	0		
		15ADON		0	0	0		
2013	China (10 provinces)	DON	50	30.0	58.1	0–862.0	0	[119]
2013	China (5 provinces)	DON	158	84.2	4084.8	23.5–25,375	68.0	[97]
		D3G		24.1	13.9	8.7–33.3		
		3ADON		84.2	14.9	10.6–177.5		
		15ADON		60.8	14.7	13.4–23.5		
		NIV		22.2	26.9	1.8–94.0		
		ZEN		77.2	85.8	13–158	24.0	
2013	Fujian	DON	59	89.8			11.2	[120]
		ZEN		11.9			0	
2013	Guangdong	DON	30	86.7	87.9	0–860.8	0	[121]
2013–2016	Shannxi	DON	504	86.9	311.0	6.0–3670.0	6.7	[122]
		3ADON		59.9	33.7	21.9–535.0		
		15ADON		8.3	5.7	4.5–105.0		
		ZEN		0.2	2.6	0–31.0	0	
2013	Tibet	DON	85	27.1	47.0	0–630.0	0	[78]
		ZEN		74.0	5.4	0–13.9	0	
2014	Xinjiang	DON	84	51.2	20.3	8.7–152.6	0	[79]
		15ADON		28.6	15.3	5.6–159.2		
2014	Henan	DON	65	69.2	218.3		0	[123]
2014	Hebei	DON	293	99.7	156.0	0–878.4	0	[124]
2014–2015	Henan	DON	295	8.8	14.4	0–750.0	0	[125]
		3ADON		0	0	0	0	
		15ADON		0	0	0	0	
		ZEN		0	0	0	0	
2016–2017	Jiangsu	DON	35	100.0	308.9	44.6–924.6	0	[126]
		3ADON		28.6	7.1	0–54.9		
		15ADON		17.1	3.2	0–23.7		
		ZEN		17.1	1.2	0–16.9	0	
		DON	50	62.0	91.9	0–401.8	0	
		3ADON		6.0	1.1	0–21.0		
		15ADON		2.0	0.3	0–14.7		
		ZEN		0	0	0	0	
2017	China	DON	75	85.3	455.7	12.5–1285.4	20.0	[127]
			15	100.0	426	51.6–1308.9	13.3	

**Table 6 toxins-11-00248-t006:** Limits of DON and ZEN relate to cereals for food and feed in China.

Food Category	Toxin	Limit (μg/kg)	Standard Code
cereal and its product: corn, corn flour (corn gluten meal, corn flake), barley, wheat, oatmeal, wheat flour	DON	1000	GB2761
plant feedstuffs		5000	GB13078
calf, lamb, concentrate supplement in lactation period		1000	
other concentrate supplement		3000	
pig formula feed		1000	
other formula feed		3000	
cereal and its product: wheat, wheat flour, corn, corn flour (corn gluten meal, corn flake)	ZEN	60	GB2761
corn and its processed products (corn bran, corn gluten feed, corn steep powder excepted)		500	GB13078
corn bran, corn gluten feed, corn steep powder, corn distiller’s grains products		1500	
other plant feedstuffs		1000	
calf, lamb, concentrate supplement in lactation period		500	
piglet formula feed		150	
gilt formula feed		100	
other pig formula feed		250	
other formula feed		500	

**Table 7 toxins-11-00248-t007:** Current detection standard for DON and ZEN by Chinese regulations.

Standard Code	Standard Category	Standard Name	Method	Toxin
GB 5009.111-2016	Mandatory national standard	Determination of deoxynivalenol and its acetylated derivatives in food	Isotope-dilution LC-MS/MS; Immunoaffinity chromatography-HPLC; thin layer chromatography	DON
GB/T 30956-2014	Recommendatory national standard	Determination of deoxynivalenol in feeds	Immunoaffinity chromatography-HPLC	
GB/T 8381.6-2005		Method for determination of deoxynivalenol in formula feed	Thin layer chromatography	
SN/T 3137-2012	Recommendatory industry standard	Determination of deoxynivalenol, 3-acety-ldeoxynivalenol, 15-O-4-acetyl-deoxynivalenol, and their metabolite in food for export	HPLC-MS/MS	
SN/T 3136-2012		Determination of aflatoxins, ochratoxin, fumonisin B1, deoxynivalenol, T-2 and HT-2 toxins in peanut, grains, and their products for export	HPLC-MS/MS	
LS/T 6110-2014	Recommendatory industry standard	Detection of deoxynivalenol in cereal	Rapid quantitative method of colloidal gold technique	
LS/T 6113-2015	Recommendatory industry standard	Detection of deoxynivalenol in grain	Rapid quantitative method of colloidal gold technique	
LS/T 6127-2017	Recommendatory industry standard	Detection of deoxynivalenol in grain	UPLC	
LS/T 6133-2018	Recommendatory industry standard	Determination of 16 mycotoxins in cereal	HPLC-MS/MS	
KJ 201702	Rapid detection standard	Rapid detection of deoxynivalenol in food	Colloidal gold immunochromatographic	
GB 5009.209-2016	Mandatory national standard	Determination of zearalenone in food	HPLC; HPLC-MS/MS; Immunoaffinity chromatography-fluorescence spectrometer	ZEN
GB/T 19540-2004	Recommendatory national standard	Determination of zearalenone in feeds	Thin layer chromatography; Enzyme-linked immunosorbent assay	
GB/T 28716-2012		Determination of zearalenone in feeds	HPLC method with immunoaffinity column clean-up	
SN/T 3235-2012	Recommendatory industry standard	Determination of multi-groups of banned drug residues in foodstuffs of Animal origin for export	LC-MS/MS	
SN/T 4058-2014	Recommendatory industry standard	Determination of residues of zeranols in foodstuffs of animal origin for export	HPLC and HPLC-MS/MS method with Immunoaffinity column clean-up	
NY/T 2071-2011	Recommendatory agriculture standard	Determination of aflatoxins, zearalenone, and T-2 in feed	LC-MS/MS	
LS/T 6109-2014	Recommendatory agriculture standard	Detection of zearalenone in cereal	Rapid method of colloidal gold technique	
LS/T 6112-2015	Recommendatory agriculture standard	Detection of zearalenone in grain	Rapid quantitative method of colloidal gold technique	
LS/T 6129-2017	Recommendatory agriculture standard	Determination of zearalenone in grains	UPLC-MS/MS	
LS/T 6133-2018	Recommendatory industry standard	Determination of 16 mycotoxins in cereal	HPLC-MS/MS	

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
