# Peer review of "Fusarium* Toxins in Chinese Wheat since the 1980s"

_toxins, 2019, doi:10.3390/toxins11050248_

Round 1

Reviewer 1 Report

After reviewing the text of the manuscript, I find that it is an interesting work that can be published after the corrections are made.

Knowledge about the occurrence of mycotoxins and diseases caused by the consumption of food contaminated with mycotoxins is extremely important. It allows recognizing the scale of this problem (the size of the risk) and taking actions to manage this risk in order to effectively strive to ensure food safety. This knowledge is even more desirable, especially in the context of climatic changes that occur in recent years, and manifested by unfavorable factors (drought, extremely high air temperature, heavy rains). The analysis of such state of knowledge over the years allows us to track the trends in the occurrence of mycotoxins, their content and draw conclusions for the future. That was missing from the manuscript. Nevertheless, this is only my observation and it is not obligatory to supplement the manuscript with these issues.

Below are other remarks:

Keywords: instead of "toxin contamination; zearalenone; deoxynivalenol; "enter:" Fusarium toxins "

Line 41-42: Give the full definition of FHB. FHB is caused not only by F. graminearum.

Line 96-97: Flour as a raw material obtained from wheat is not eaten by people, only after its processing. Change this sentence. The same is true for sentence in line 98-99.

Line 123-125: In many places around the world, it was shown that the incidence of esophageal cancer is related to grain (maize) fumonisins contamination. From my own experience I know that fumonisins are found not only in corn, but also rice. What is the consumption of these cereals (intended for consumption) by the population living in these areas. Is there any data? I think that the incidence of esophageal cancer is not only the result of the presence of DON or NIV in cereal grains, but also other mycotoxins. I think that at the end of the paragraph you can add your own thoughts (conclusions) about the causes of this disease.

Line 165: Instead of F. graminearum species complex (FGSC) use FGSC - the abbreviation was already before (line 42)

Line 174-175: Tables, e.g. Table 2, should be under the paragraph under which it was first mentioned.

Line 229: (detail) instead of 577 should be 578 μg / kg the same in line 234 instead of 4501 should be 4502 μg / kg. Throughout the text, authors should agree on the method of recording. Once the authors round the values and once not. This should be harmonized.

Line 237: The text is for the first time about modified derivatives of DON. The authors should present in a few sentences how they are found in wheat grain and present its toxicity in comparison to free DON (in the paragraph where the toxicity of trichothecenes is mentioned).

Line 250-255: should be moved to the paragraph in which the toxicity of trichothecenes is described.

Line 288-292: Please,add reference.

"Maturation of chemical control measure of FHB" - this section presents potential possibilities of fungicides. This chapter is well written, but it must be emphasized that the world is currently seeking to limit the use of these substances, as they are themselves a threat to food safety as they are remnants of pesticides found in food.

"Conclusions and challenges for the future" - the conclusions are factual and concrete. Most of these factors limiting the development of FHB, and consequently mycotoxins, have been known for a very long time. In this chapter, I would lean over new issues related to ensuring food safety. For example, work on wheat varieties resistant to FHB, which in the face of global warming (and as a consequence of adverse weather events causing stress in plants and other organisms (fungi), such as drought) will be able to cope better in stressful situations. If the authors decide to mention these things, it would be appropriate to perform the characteristics of current research in this area in the relevant chapter and to draw conclusions for the future.

Author Response

Reviewer 1:

1. Keywords: instead of "toxin contamination; zearalenone; deoxynivalenol; "enter:" Fusarium toxins”

    We have made the adjustment.

2. Line 41-42: Give the full definition of FHB. FHB is caused not only by F. graminearum.

    We have modified the statement.

3. Line 96-97: Flour as a raw material obtained from wheat is not eaten by people, only after its processing. Change this sentence. The same is true for sentence in line 98-99.

    We have changed these two sentences.

4. Line 123-125: In many places around the world, it was shown that the incidence of esophageal cancer is related to grain (maize) fumonisins contamination. From my own experience I know that fumonisins are found not only in corn, but also rice. What is the consumption of these cereals (intended for consumption) by the population living in these areas. Is there any data? I think that the incidence of esophageal cancer is not only the result of the presence of DON or NIV in cereal grains, but also other mycotoxins. I think that at the end of the paragraph you can add your own thoughts (conclusions) about the causes of this disease.

We agree with your opinion. The occurrence of esophageal cancer is associated with various mycotoxins. In this text, we focus the relationship between the incidence and the toxin in wheat grains from the few report. We have added several sentences about the causes of this disease at the end of the paragraph.

5. Line 165: Instead of F. graminearum species complex (FGSC) use FGSC - the abbreviation was already before (line 42)

The explanation of FGSC has been deleted.

6. Line 174-175: Tables, e.g. Table 2, should be under the paragraph under which it was first mentioned.

We have put the tables in the corresponding position.

7. Line 229: (detail) instead of 577 should be 578 μg / kg the same in line 234 instead of 4501 should be 4502 μg / kg. Throughout the text, authors should agree on the method of recording. Once the authors round the values and once not. This should be harmonized.

Toxin concentration in this article was picked in the original paper and we did not make any change. At present, a decimal portion is reserved in all figures.

8. Line 237: The text is for the first time about modified derivatives of DON. The authors should present in a few sentences how they are found in wheat grain and present its toxicity in comparison to free DON (in the paragraph where the toxicity of trichothecenes is mentioned).

    We have added some sentences about the first discovery and toxicity of D3G in this paragraph.

9. Line 250-255: should be moved to the paragraph in which the toxicity of trichothecenes is described.

    We suggest that these sentences keep in former position. DON and ZEN are the main research content in this article and there is not separate explanation of acetyl DON. In addition, the toxicity of D3G is still n clear.

10. Line 288-292: Please,add reference.

We have added the necessary references.

11. "Maturation of chemical control measure of FHB" - this section presents potential possibilities of fungicides. This chapter is well written, but it must be emphasized that the world is currently seeking to limit the use of these substances, as they are themselves a threat to food safety as they are remnants of pesticides found in food.

    We have added relevant content in the paragraph.

Reviewer 2 Report

A very interesting and well written review.

I have the following minor suggestions:

1) a list af abbreviations would be helpful

2) Tables: check formatting, in tables covering more than one page, the first row should be shown on every page

3) L256 multitoxins?

Author Response

Reviewer 2:

1. a list of abbreviations would be helpful

We consider this is a very good suggestion. The list of abbreviations seems to be uncommon in this journal and all abbreviations have been defined in the first time they appear in the text according to the policy.

2. Tables: check formatting, in tables covering more than one page, the first row should be shown on every page

    We have adjusted table format and the first row of the tables covering more than one page has shown on every page.

3. L256 multitoxins?

    We have replaced “multitoxins” with “multitoxin” in the article.

Reviewer 3 Report

This is an interesting and valuable paper, a source of important information. It is well written and I have no major comments to it, just a few suggestions: 

Line 165: no need to explain the abbreviation FGSC, it is already explained in the line 42. 

It is not usual to put tables at the end of the conclusion. I suggest a better incorporation of these tables into the text body. 

Some tables have units marked in red. Please correct.  

Tables 4-7 are not mentioned in the text. Please add a sentence about these tables into the text. 

Author Response

Reviewer 3:

1. Line 165: no need to explain the abbreviation FGSC, it is already explained in the line 42.

    The explanation of FGSC has been deleted.

2. It is not usual to put tables at the end of the conclusion. I suggest a better incorporation of these tables into the text body.

We have put the tables in the corresponding position.

3. Some tables have units marked in red. Please correct.               

It is our fault and we have corrected the mistakes.

4. Tables 4-7 are not mentioned in the text. Please add a sentence about these tables into the text.

Tables 4 and 5, showing the main Fusarium toxins in wheat grains and wheat flour, existed in the article and we have added corresponding sentences about tables 6 and 7 in the text.

Reviewer 4 Report

This review about Fusarium toxins in China since 1980s is very important contribution to this field. It is very well organized and I do not have some important comments.

Only few minor things, for examplr  in line 166, 173, 177 I do not know why are you using 's.str.' after Fusarium species.

Also, in some caces you are using decimal points for toxin concentration in some not.

Author Response

Reviewer 4:

1. Only few minor things, for example in line 166, 173, 177 I do not know why are you using 's.str.' after Fusarium species.

Traditional and previous Fusarium graminearum is determined Fusarium graminearum species complex after long term studies on population genetics. Fusarium graminearum s. str. refers specifically to the lineage 7 in FGSC, distributing all over the world; while Fusarium asiaticum, mainly occurred in Asia, is lineage 6 in FGSC. In order to distinguish between FGSC and lineage 7, 's.str.' is added.

2. In some cases you are using decimal points for toxin concentration in some not.

Toxin concentration in this article was picked in the original paper and we did not make any change. At present, a decimal portion is reserved in all figures.

Round 2

Reviewer 1 Report

This version of the manuscript has been modified and is more acceptable.